# Nono induces Gadd45b to mediate DNA repair

Victoria Mamontova[1,2,*], Barbara Trifault[1,2,*], Kaspar Burger[1,2]

**RNA-binding proteins are frequently deregulated in cancer and emerge as effectors of the DNA damage response (DDR). The non-POU domain–containing octamer-binding protein NONO/p54[nrb] is a multifunctional RNA-binding protein that not only modulates the production and processing of mRNA, but also promotes the repair of DNA double-strand breaks (DSBs). Here, we investigate the impact of _Nono_ deletion in the murine KP (_KRas^{G12D}_, _Trp53^{−/−}_) cell–based lung cancer model. We show that the deletion of Nono impairs the response to DNA damage induced by the topoisomerase II inhibitor etoposide or the radiomimetic drug bleomycin. Nono-deficient KP (KPN) cells display hyperactivation of DSB signalling and high levels of DSBs. The defects in the DDR are accompanied by reduced RNA polymerase II promoter occupancy, impaired nascent RNA synthesis, and attenuated induction of the DDR factor growth arrest and DNA damage–inducible beta (Gadd45b). Our data characterise Gadd45b as a putative Nono-dependent effector of the DDR and suggest that Nono mediates a genome-protective crosstalk of the DDR with the RNA metabolism via induction of Gadd45b.**

## Introduction

The DNA damage response (DDR) repairs DNA double-strand breaks (DSBs), which are toxic and threaten genome stability (Jackson & Bartek, 2009; Ciccia & Elledge, 2010). The recognition of DSBs is governed by kinases such as ataxia–telangiectasia mutated and DNA-activated protein kinase (DNA-PK), which phosphorylate >100 DDR factors to facilitate DSB repair (DSBR) (Kastan & Lim, 2000; Blackford & Jackson, 2017). Intriguingly, about 40% of such phosphorylations address factors related to nucleic acid metabolism, in particular RNA-binding proteins (RBPs) (Dutertre et al, 2014; Burger et al, 2019). Phosphorylation of the endoribonuclease DICER, for instance, activates the processing of DNA damage–induced transcripts to promote the recruitment of DSBR factors such as the p53-binding protein 1 (53BP1) to DSBs (Francia et al, 2012; Wei et al, 2012; Burger et al, 2017; Burger & Gullerova, 2018). DSBR also occurs via RNA-templated DNA repair or uses non-coding transcripts to scaffold the recruitment of repair factors (Keskin et al, 2014;

Chakraborty et al, 2016). Thus, various modes of RNA-dependent DSBR coexist and stimulate canonical DSBR.

Genome instability is a hallmark of many tumours, including lung cancer (Tubbs & Nussenzweig, 2017). Lung cancer is classified into small-cell lung cancer (SCLC) and non–small-cell lung cancer (NSCLC) (Travis et al, 2015). NSCLCs comprise a high frequency of somatic mutations that include not only bona fide tumour drives (Kandoth et al, 2013; Bielski et al, 2018; Satpathy et al, 2021), but also DDR factors such as ataxia–telangiectasia mutated (Ding et al, 2008). The high mutational burden in lung cancer is often accompanied by the development of resistances that interfere with the targeted therapy (Lim & Ma, 2019). Thus, lung cancer remains an aggressive disease with poor prognosis for patients.

Challenges in lung cancer treatment underscore the relevance of identifying novel vulnerability. Interestingly, the RNA metabolism may represent such vulnerability (Abdel-Wahab & Gebauer, 2018). Indeed, 42 of 723 genes that contribute to tumorigenesis encode RBPs (Forbes et al, 2015; Choi & Thomas-Tikhonenko, 2021). Strikingly, many tumour-promoting RBPs are components of paraspeckles and some of them are classified as drivers of neoplastic growth (Cerami et al, 2012; Naganuma et al, 2012). The multifunctional non-POU domain–containing octamer-binding protein NONO/p54[nrb] is a core component of paraspeckles, which are stress-responsive bimolecular condensates that form in the interchromatin space of mammalian nuclei (Knott et al, 2016; Fox et al, 2018). In addition, NONO dynamically associates with sites of active transcription to modulate RNA polymerase II (RNAPII) activity, which drives the expression of oncogenic transcriptional programs in many tumours, including lung cancer (Chen et al, 2016; Feng et al, 2020; Wei et al, 2021). Interestingly, NONO is also linked to genome maintenance. NONO promotes the activity of DNA-PK via its ability to phase-separate and stimulates survival upon ionising irradiation (Krietsch et al, 2012; Wang et al, 2022). The depletion of NONO, in turn, causes genome instability (Li et al, 2009; Petti et al, 2019). However, little is known about the genome-protective role of murine Nono in NSCLCs. Here, we use murine KP (_KRas^{G12D}_, _Trp53^{−/−}_) cells—a CRISPR-mediated cell-based NSCLC system that is comparable to the classic Cre-mediated KP mouse model system (_KRas^{G12D}_, _Trp53^{fl/fl}_) (Hartmann et al, 2021)—to investigate the impact of _Nono_ deletion on genome integrity. We show that _Nono_ deletion hypersensitises KP cells to DNA damage and postulate that

[1]Mildred Scheel Early Career Center for Cancer Research (Mildred-Scheel-Nachwuchszentrum, MSNZ) Würzburg, University Hospital Würzburg, Würzburg, Germany [2]Department of Biochemistry and Molecular Biology, Biocenter of the University of Würzburg, Würzburg, Germany

Correspondence: kaspar.burger@uni-wuerzburg.de
*Victoria Mamontova and Barbara Trifault contributed equally to this work

Nono mitigates the accumulation of DNA damage by promoting the expression of the DDR factor growth arrest and DNA damage–inducible beta (Gadd45b).

# Results and Discussion

### *Nono* deletion impairs RNAPII activity upon DNA damage

NONO stimulates the formation of transcriptionally active bimolecular condensates in vitro and colocalises with RNAPII in vivo (Lewis et al, 2023; Zhang et al, 2023). To approach the role of murine Nono in KP lung cancer cells, we used the CRISPR/Cas9 technology and created the monoclonal *Nono* knockout cell line KPN. The absence of Nono expression was confirmed by immunoblotting and confocal imaging (Fig S1A and B). We wished to test whether *Nono* deletion alters RNAPII activity in response to DNA damage. We incubated KP or KPN cells in the presence or absence of the topoisomerase II inhibitor etoposide, and assessed RNA synthesis globally by confocal imaging of nascent transcripts that were pulse-labelled with 5-ethynyl uridine (EU) (Fig 1A). We observed a partial reduction in EU signals upon etoposide treatment in KP cells, which was more distinct in KPN cells. Next, we investigated RNAPII promoter occupancy and performed CUT&RUN-seq with an antibody that selectively recognises serine-5 phosphorylated residues of the RNAPII carboxy-terminal domain (CTD S5P). Indeed, combining *Nono* deletion with etoposide treatment significantly reduced CTD S5P occupancy at highly expressed genes, but had no impact in untreated cells (Fig 1B). This phenotype was confirmed by visual inspection of browser tracks at a subset of highly expressed genes like *Actb* (Fig 1C). To validate CUT&RUN-seq data, we performed manual chromatin immunoprecipitation (ChIP) and confirmed a partial loss of CTD S5P occupancy at the *Actb* promoter region upon combining *Nono* deletion with etoposide treatment (Fig 1D). Thus, Nono stimulates RNAPII activity and CTD S5P promoter occupancy upon etoposide treatment.

### *Nono* deletion triggers genome instability and impairs DNA repair

We could recently show that the DDR triggers NONO nucleolar relocalisation to detain aberrant, intron-containing pre-mRNA transcripts, thereby promoting genome stability in U2OS cells upon etoposide treatment (Trifault et al, 2022, 2024). To assess whether the etoposide-induced defects in RNAPII activity and promoter occupancy in Nono-deficient KP cells correlate with genome instability, we compared the amount of broken chromatin in KP and KPN cells upon treatment with etoposide and other cytotoxic drugs by neutral comet assays (Fig 2A). Although etoposide treatment or *Nono* deletion alone induced modest appearance of DNA tails, we found more prominent DNA damage in a subset of etoposide-treated KPN cells. Nono-deficient KP cells were also hypersensitive to treatment with the radiomimetic drug bleomycin, but not to treatment with the DNA interstrand cross-linker cisplatin. Next, we applied DNA damage in situ ligation followed by the proximity ligation assay (DI-PLA), a method that uses ligation of a biotinylated DNA linker to blunted DSBs to detect persistent DNA damage as PLA foci in single cells (Galbiati et al, 2017). To perform DI-PLAs, we combined a biotin antibody with an antibody that recognises the DNA damage marker ser-139 phosphorylated histone H2A.X variant (γH2A.X), and quantified the number of dots per nucleus by confocal imaging (Fig 2B). Stratification revealed that the treatment with etoposide significantly increased the number of cells that were positive for DI-PLA signals, but reduced the number of KP cells without DI-PLA dots. Strikingly, this phenotype was elevated in Nono-deficient KP cells. We conclude that the deletion of *Nono* hypersensitises KP cells to DSB-inducing drugs.

Next, we wished to test the impact of *Nono* deletion on the recognition and repair of DSBs. We used etoposide incubation kinetics and compared the level of γH2A.X signals in KP and KPN cells (Fig 3A). We found that *Nono* deletion increased the amount of etoposide-induced γH2A.X levels twofold to threefold and also delayed the clearance of γH2A.X upon chase. The delayed clearance of γH2A.X levels could also be confirmed upon chasing bleomycin treatment (Fig 3B). Importantly, γH2A.X levels could partially be rescued by complementation with ectopically expressed mCherry-tagged human NONO (Fig 3C and D). Next, we used confocal imaging of the DSB marker 53BP1 to compare the formation of etoposide-induced DSB foci in KP and KPN cells. We observed that the number of KPN cells that displayed persistent 53BP1-positive staining was significantly increased in KPN cells upon chasing etoposide treatment (Fig 3E). A similar phenotype was detected upon staining for γH2A.X-positive foci, which were elevated in KPN cells both after etoposide incubation and chase (Fig 3F). We conclude that Nono promotes efficient DSBR in KP cells.

### Nono-dependent induction of Gadd45b promotes the DDR

Our data point towards defects in RNAPII transcription that are prevalent in DNA-damaged Nono-deficient KP cells and correlate with inefficient DSBR. To identify genes that may suppress such defects in Nono-proficient cells, we performed RNA-seq in KP and KPN cells in the absence or presence of etoposide. We first asked whether etoposide treatment induces gene sets in KP cells that are sensitive to *Nono* deletion. We identified gene sets involved in TNF-α signalling, the UV response, and the p53 pathway as highly significantly up-regulated upon etoposide treatment in both KP and KPN cells (Fig S2A and B). When scoring the relative induction of these three gene sets in etoposide-treated KPN cells relative to etoposide-treated KP cells, we found that the DNA damage–induced expression of the TNF-α gene set was selectively impaired in etoposide-treated KPN cells. Importantly, RNA-seq data were highly reproducible and consistent among replicates (Fig S2C). Next, we visualised the relative abundance of mRNA transcripts in KP and KPN cells in response to etoposide treatment on volcano plots (Fig 4A). In total, we identified 162 candidate mRNA transcripts that were significantly elevated in levels by etoposide in KP cells. Strikingly, 5 mRNA transcripts that belong to the TNF-α signalling gene set (*Lif*, *Dusp1*, *Gadd45b*, *Sgk1*, and *Phlda1*) were found among the top 20 induced candidates and all 5 transcripts failed to be induced by etoposide in KPN cells. Among the five candidates, *Gadd45b* caught our attention as it represents a prognostic marker in lung cancer and encodes a nuclear protein that modulates both RNAPII activity

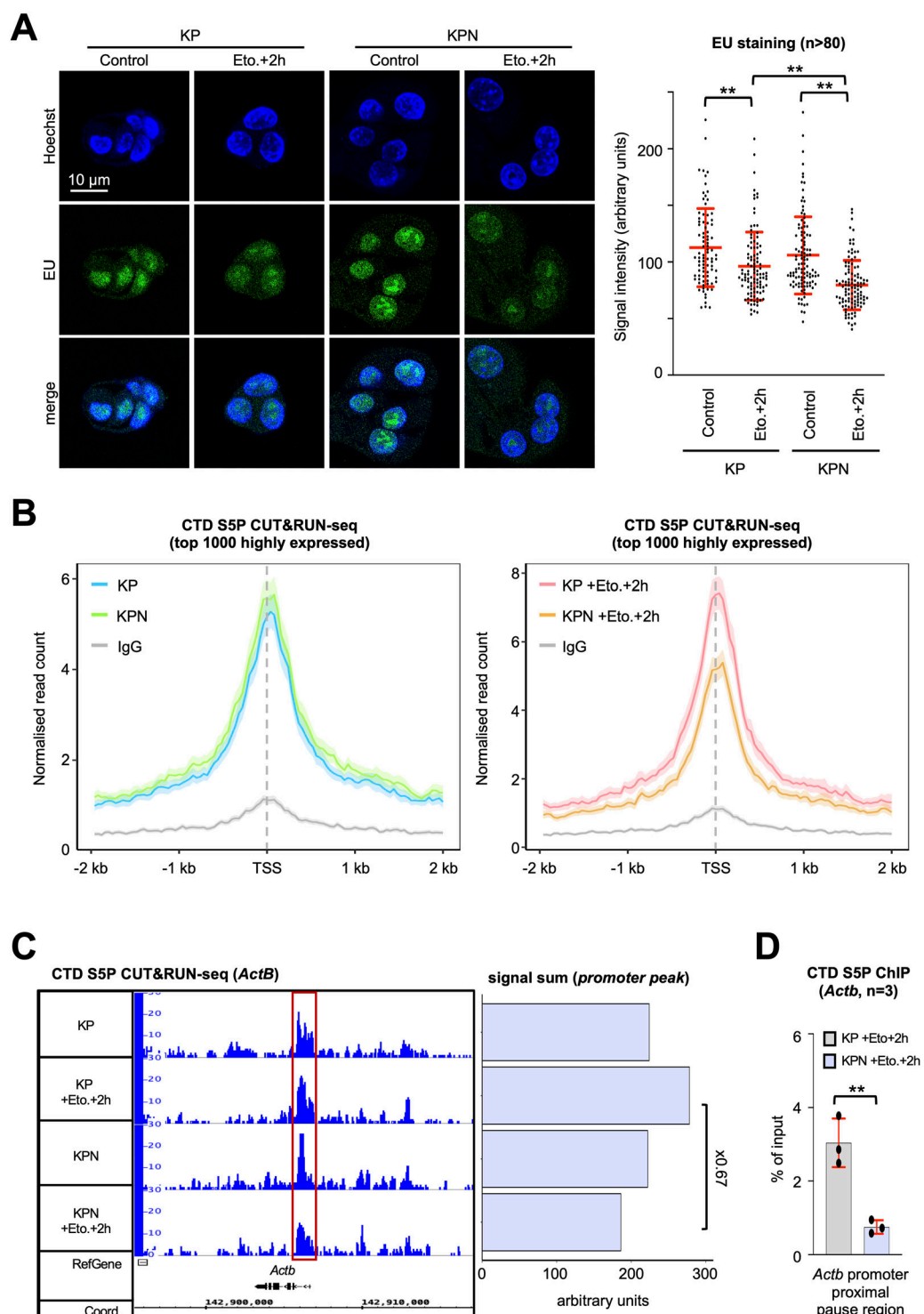

**Figure 1. Defective RNAPII activity upon etoposide treatment in Nono-deficient KP cells.**
**(A)** Imaging (left) and quantitation (right) of 5-ethynyl uridine (EU)–labelled nascent transcripts. n, number of cells. **(B)** CTD S5P CUT&RUN-seq metagene plots for transcription start sites (TSSs) of the top 1,000 highly expressed genes in the absence or presence of Nono (left) or upon etoposide incubation (right) IgG, immunoglobulin, background. **(C)** Browser tracks (left) and quantitation (right) depicting CTD S5P CUT&RUN-seq reads at the *Actb* locus. Red box, promoter area. **(D)** Manual ChIP detecting CTD S5P occupancy at the *Actb* locus using site-specific primers. n, number of biological replicates. */**, *P*-value < 0.05/<0.001; two-tailed *t* test. Error bar, mean ± SD. Source data are available for this figure.

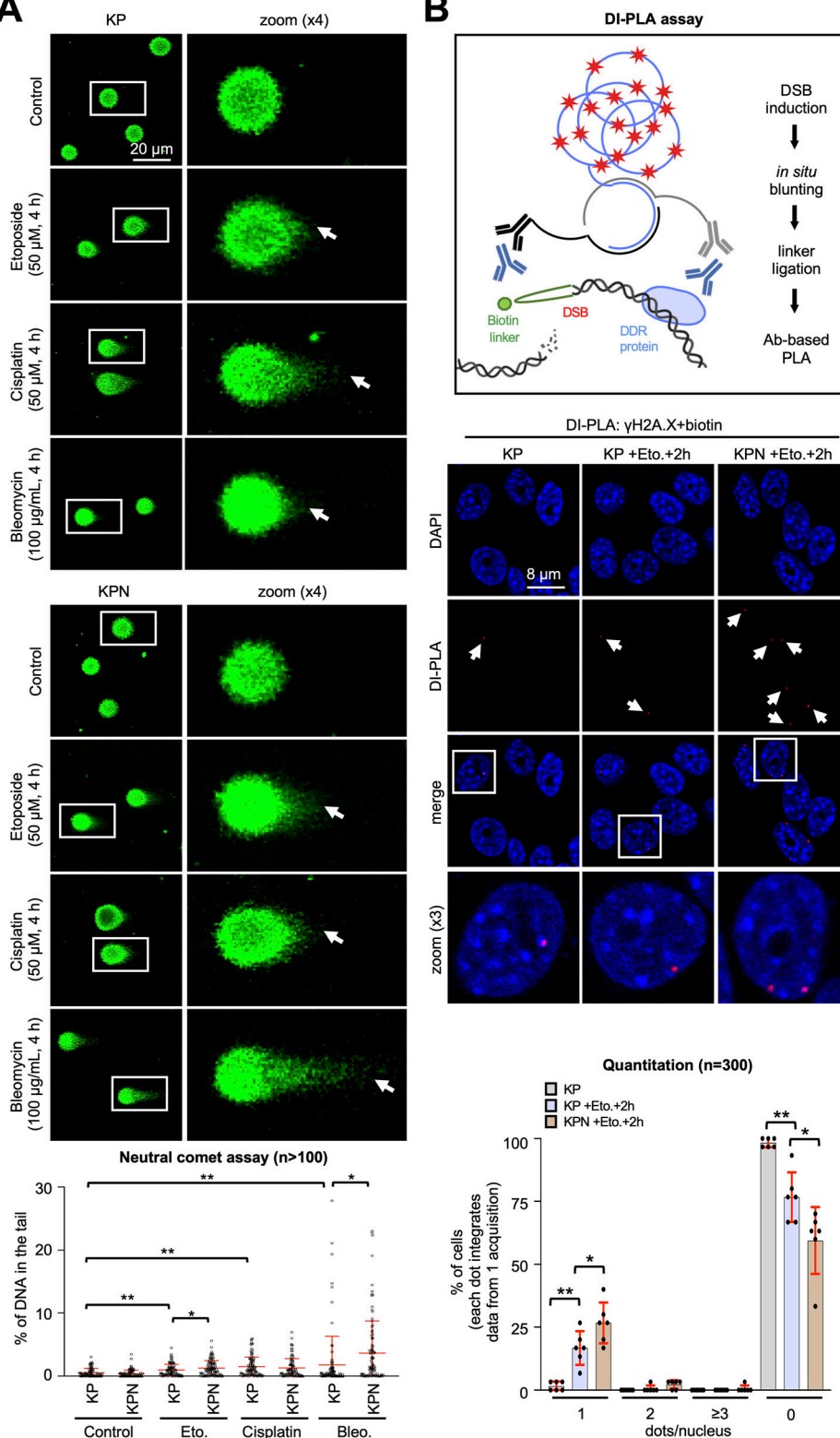

Figure 2. Elevated DNA damage in Nono-deficient KP cells.
(A) Imaging (top) and quantitation (bottom) of DNA subjected to a neutral comet assay. White box, zoom; arrowhead, DNA tail; n, number of cells.
(B) Scheme (top), imaging (middle), and quantitation (bottom) of DI-PLA signals obtained from costaining with a ser-139 phosphorylated histone H2A.X variant (γH2A.X) and biotin antibodies. White box, zoom; arrowhead, DI-PLA signal; black box, scheme of the DI-PLA; n, number of cells. */**, P-value < 0.05/<0.001; two-tailed t test. Error bar, mean ± SD.
Source data are available for this figure.

and the DDR (Niehrs & Schäfer, 2012; Lv et al, 2023). We validated the DNA damage–responsive onset of Gadd45b expression upon etoposide treatment in KP cells and its attenuated induction in KPN

cells by RT–qPCR, confocal imaging, and immunoblotting (Figs 4B and C and S2D). Importantly, the attenuated Gadd45b transcript and protein expression in KPN cells could also be observed upon

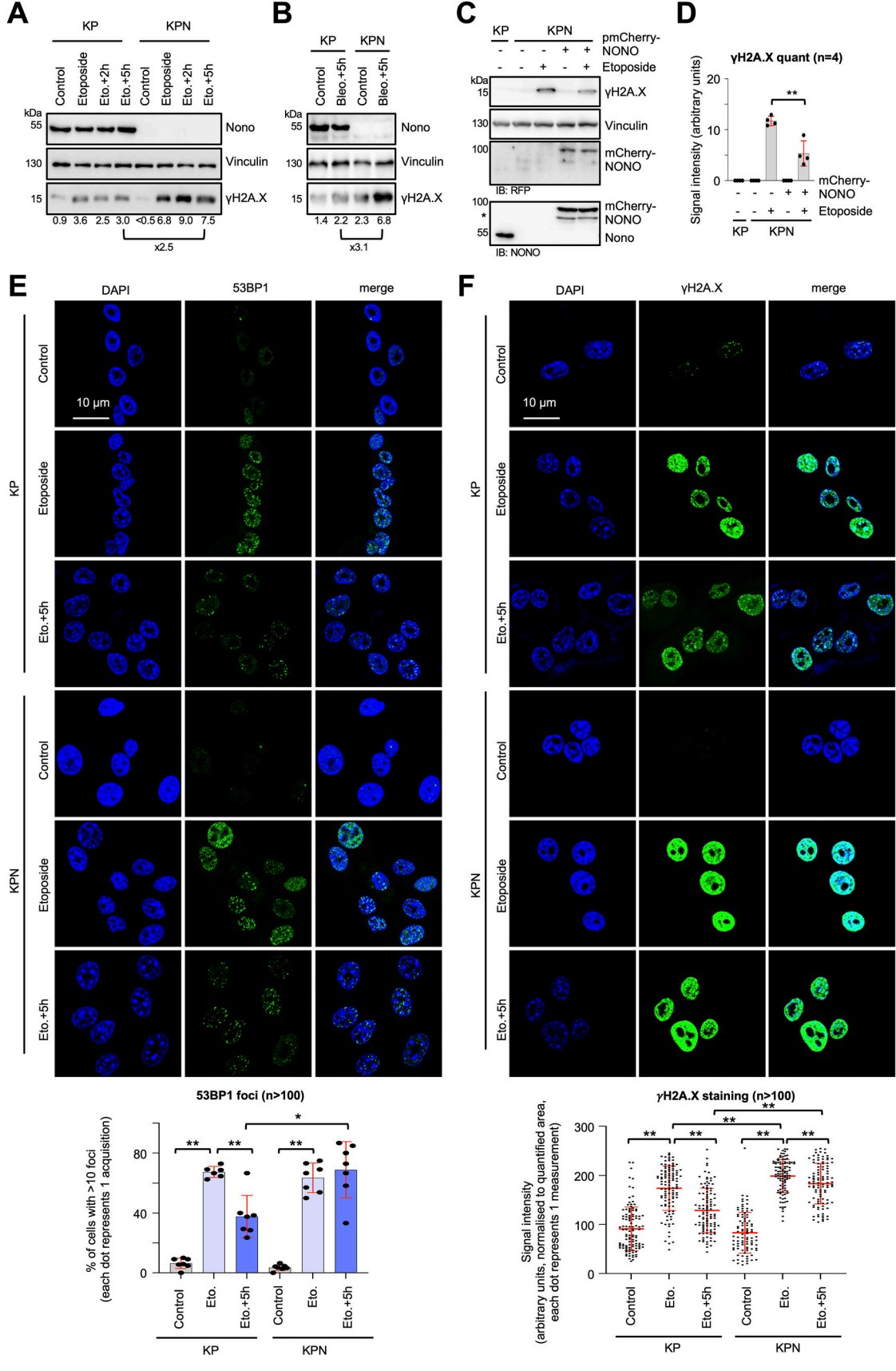

treatment with bleomycin (Figs 4D and S2E). Next, we asked whether the underrepresentation of Gadd45b mRNA transcripts in etoposide-treated KPN cells is reflected in CTD S5P promoter occupancy. We reassessed our CUT&RUN-seq data, which indeed displayed an about twofold reduction in CTD S5P occupancy both at the *Gadd45b* promoter and at the promoters of 3 of the other 4 TNF-*α* signalling genes upon combining etoposide treatment with *Nono* deletion (Fig S3A–C). Reassuringly, etoposide treatment combined with *Nono* deletion did not severely alter CTD S5P occupancy at the promoter region of at least five other RNA-seq candidates, which were induced by etoposide treatment irrespective of *Nono* deletion. To test whether the Nono-dependent expression of Gadd45b modulates the DDR, we preincubated KP or KPN cells with the selective Gadd45b inhibitor DTP3, which prevents Gadd45b from binding to the stress-responsive mitogen-activated protein kinase kinase (MAPKK) MKK7, thereby promoting efficient MAPKK signalling (Tornatore et al, 2014). We assessed the level of etoposide-induced γH2A.X signals by immunoblotting and confocal imaging (Fig 5A and B). Indeed, combining etoposide treatment with DTP3 elevated γH2A.X signals in both KP and KPN cells in a dose-dependent manner and appeared additive in KPN cells. We also assessed the CTD S5P phosphorylation status of RNAPII by immunoblotting and observed that neither *Nono* deletion nor DTP3 treatment altered the amount of total RNAPII or CTD S5P marks upon etoposide treatment (Fig S4A). Importantly, treatment with a high concentration of DTP3 (20 μM) elevated the levels of phosphorylated c-Jun N-terminal kinase (p-JNK) upon etoposide treatment, suggesting that the DTP3 inhibitor is active in both KP and KPN cells (Fig S4B). Finally, we wished to test the impact of the Gadd45b overexpression of genome stability in KP and KPN cells. Using a neutral comet assay and immunoblotting, we found that the overexpression of human HA-tagged GADD45 attenuates the excessive formation of DNA damage upon bleomycin treatment in Nono-deficient KP cells (Figs 5C and D and S4C). We conclude that Nono promotes the expression of Gadd45b to stimulate DSB signalling in KP cells.

We report defects in RNA metabolism upon DNA damage in Nono-deficient KP cells and impaired DNA repair that are associated with defective Gadd45b expression in KPN cells (Fig 5E). How are Nono and Gadd45b linked? Nono associates with chromatin to stimulate both RNAPII activity and the formation of CTD S5P marks at a subset of protein-coding genes that are critical for the differentiation of mouse embryonic stem cells (Ma et al, 2016). Likewise, Gadd45b regulates the mitogen-activated protein kinase pathway, for instance, by activating the p38 kinase, which is also required for the modulation of the RNA metabolism upon DNA damage in human tissue culture cells (Yoo et al, 2003; Borisova et al, 2018). Perturbations in the activity of RNAPII can impact on premRNA processing and, for instance, trigger the accumulation of aberrant transcripts such as DNA-RNA hybrids (R-loops) (de la Mata et al, 2003; Castillo-Guzman et al, 2020 *Preprint*). High levels of R-loops correlate with genome instability (García-Muse & Aguilera,

2019; Marnef & Legube, 2021). Interestingly, chromatin binding of Gadd45a is R-loop–dependent and the binding sites of human NONO to chromatin also strongly correlate with R-loop–prone sites (Arab et al, 2019; Wu et al, 2022). We postulate that murine Nono induces Gadd45b upon DNA damage to mitigate the formation of R-loops and that Nono and Gadd45b perhaps modulate both RNAPII activity and R-loop levels upon DNA damage in a shared pathway. However, the precise mechanism remains elusive. Of note, human NONO regulates gene expression also at the posttranscriptional level, for instance, by promoting the retention and editing of transcripts (Knott et al, 2016). Thus, we cannot exclude that the posttranscriptional roles of Nono are compromised in KPN cells and contribute to the phenotypes observed in this study.

Our data predict that the inactivation of Nono in KP cells may trigger genome instability upon chemotherapy and thus provide vulnerability in lung cancer. One option may be to interfere with the affinity of Nono to bind chromatin-associated nascent RNA. A panel of electrophilic small molecules has recently been described as direct modulators of human NONO RNA-binding affinity. (R)-SKBG-1 and other NONO ligands react with the cysteine-145 residue of NONO, which stabilises NONO RNA interactions and triggers NONO mislocalisation in nuclear foci that are accompanied by diminished oncogenic gene expression and growth defects in prostate cancer cells (Kathman et al, 2023). Interestingly, blockage of the protein arginine methyltransferase PRMT4/CARM1 with the small molecule inhibitor EZM-2302 has recently been established as novel vulnerability in tumours (Kumar et al, 2021). PRMT4/CARM1 methylates both the RNAPII CTD and the coiled-coil domain of NONO to stimulate RNAPII activity and attenuate the binding affinity of NONO to RNA, respectively (Sims et al, 2011; Hu et al, 2015). Thus, the growth defects observed upon EZM-2302 treatment may, at least in part, be caused by interference with NONO function. It is tempting to speculate that the combination of small molecule inhibitors with genotoxic stress inactivates Nono and triggers RNA-induced toxicity in KP cells. To understand the physiological relevance of the Nono-mediated DDR proposed in this study, it will be important to test the capability of KPN cells to form tumours in vivo and assess the sensitivity of such tumours to genotoxic stress in the future.

## Materials and Methods

### Tissue culture and transfections

Murine KP (*KRas^{G12D}*, *Trp53^{-/-}*) cells, KP-derived *Nono* knockout KPN (*KRas^{G12D}*, *Trp53^{-/-}*, *Nono^{-/-}*) cells, and human HEK293T cells were cultured in DMEM (Gibco) with 10% FBS (Capricorn), 100 U/ml penicillin–streptomycin (Gibco), and 2 mM L-glutamine (Gibco) at 37°C and 5% $CO_2$. Cells were incubated with etoposide (20 μM or as indicated; Sigma-Aldrich), bleomycin (100 μg/ml; Hycultec), and cisplatin (50 μM; Sigma-Aldrich) for 2–4 h or chase times or

**Figure 3. Hyperactive DSB signalling upon etoposide treatment in Nono-deficient KP cells.**
**(A, B, C)** Immunoblots detecting Nono and γH2A.X from whole-cell lysates upon treatment with etoposide (A) or bleomycin (B), or upon transient transfection of mCherry-tagged human NONO (C). Vinculin, loading control; asterisk, unspecific; IB, immunoblot. **(C, D)** Quantitation of γH2A.X levels from (C). n, number of biological replicates. **(E, F)** Imaging (top) and quantitation (bottom) of 53BP1 (E) or γH2A.X (F). n, number of cells. */**, *P*-value < 0.05/<0.001; two-tailed *t* test. Error bar, mean ± SD. Source data are available for this figure.

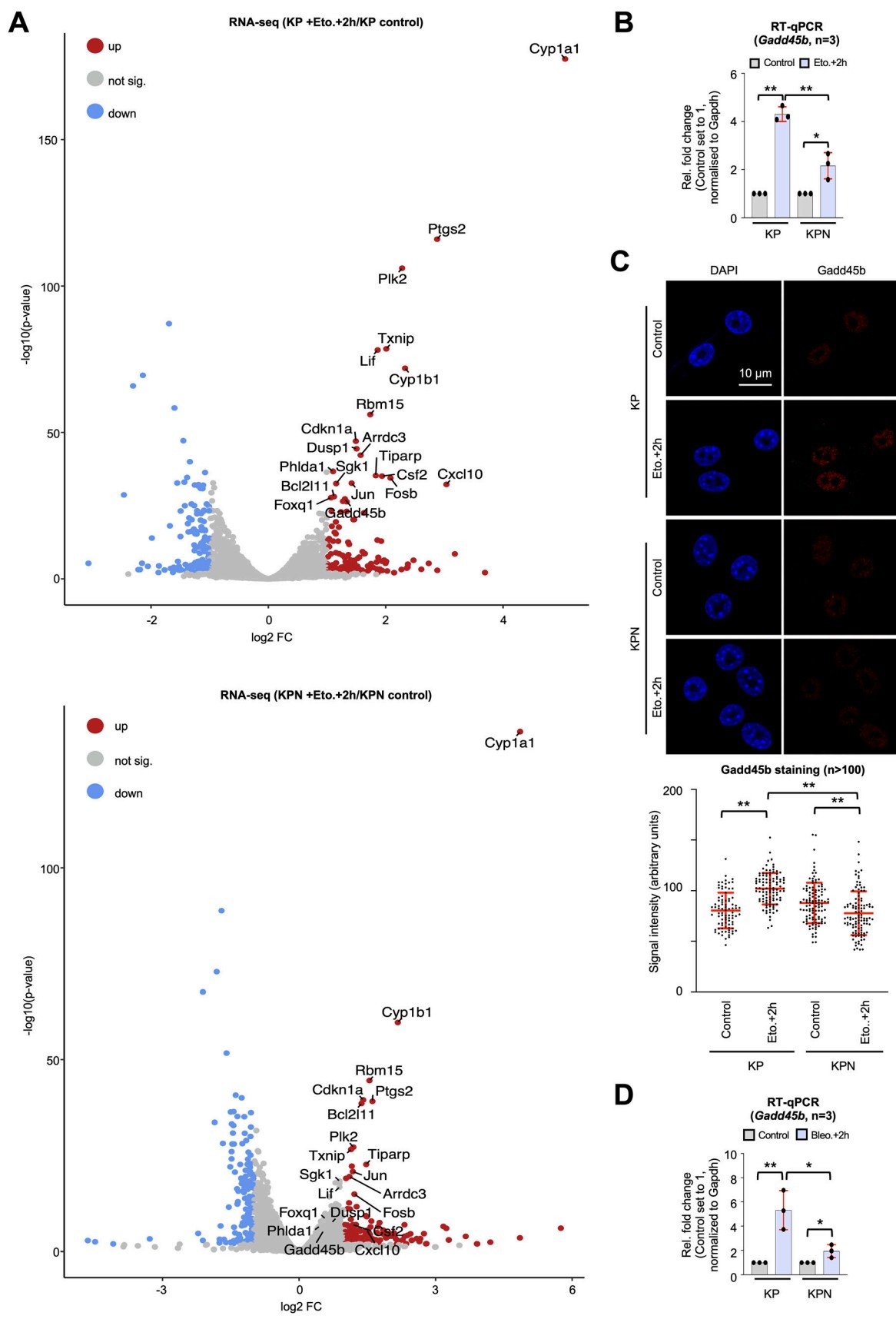

preincubated with the Gadd45b inhibitor DTP3 (Selleckchem) as indicated. Transfection of plasmids pmCherry-NONO (a kind gift from Ling-Ling Chen) and pcDNA3-HA-WT-GADD45 (Addgene) was performed using Lipofectamine 2000 (Invitrogen) and Opti-MEM (Gibco) following the manufacturer's protocol.

### CRISPR cloning and lentiviral transduction

For cloning of the Nono-targeting CRISPR vector, the parental lentiCRISPR_v2 plasmid (a kind gift from Markus Diefenbacher) was restricted with BsmBI-v2 (NEB) and ligated with either of the two preannealed sgRNA-encoding DNA oligonucleotide inserts, which were designed using the CHOPCHOP online tool (Table S1). The correct integration of sgRNA sequences into pCRISPR-V2-sgRNA-*Nono*-targeting vectors was validated by Sanger sequencing. To generate monoclonal *Nono* knockout cells, 10 μg pCRISPR-V2-sgRNA-Nono plasmid was pooled with psPAX2 and pMD7.G (kind gifts from Elmar Wolf), mixed with 30 μl polyethylenimine (Calbiochem), diluted in 500 μl Opti-MEM, vortexed, incubated (25 min, RT), added to HEK293 cells that were preincubated in 5 ml DMEM/2% FBS, and transfected (8 h). The virus was harvested two times every 12 h, sterile-filtered, and frozen. For infection, KP cells were cultured (24 h) in viral mixture (1.5 ml DMEM, 1.5 ml viral harvest, 6 μl polybrene; Invitrogen). The mixture was replaced by DMEM with 2.5 μg/ml puromycin (Invitrogen) for polyclonal selection (10 d). Individual Nono knockout candidate clones were yielded by single-cell suspension and validated by downstream assays.

### Immunoblotting

Proteins were assessed as whole-cell extracts that were directly lysed, boiled, and sonicated in 4x sample buffer (250 mM Tris–HCl, pH 6.8, 8% SDS, 40% glycerol, 0.8% $\beta$-mercaptoethanol, 0.02% bromophenol blue). Samples were separated by SDS–PAGE and transferred to nitrocellulose membranes (Cytiva), blocked and washed in PBS/0.1% Triton x-100/5% milk (PBST), probed with selective antibodies (Table S2), and visualised with an ECL kit (Cytiva) and an imaging station (LAS-4000, Fuji; Fusion FX, Vilber). Signals were quantified by ImageJ (NIH).

### Indirect immunofluorescence

Cells were grown on coverslips (Roth), washed in PBS, fixed (10 min) in 3% paraformaldehyde (Sigma-Aldrich), washed in PBS (three times, 5 min), permeabilised with PBS/0.1% Triton x-100 (10 min), and blocked with PBS/10% FBS (2 h, 4°C). Primary and secondary antibodies (Table S2) were diluted in PBS/0.15% FBS and incubated in a humidified chamber (overnight, 4°C, or 2 h, RT), respectively. Cells were washed between incubations with PBS/0.1% Triton x-100 (three times, 5 min), sealed in DAPI-containing mounting medium

(Vectashield), and imaged by confocal microscopy (CLSM; Leica SP2, 1,024 × 1,024 resolution, 63x, airy = 1). Channels were acquired as single snapshots, sequentially, between frames, with equal exposure times. Data were analysed using ImageJ (NIH). At least 80 cells per condition were quantified. For imaging of nascent RNA, cells were incubated with 1 ml DMEM containing 1 mM EU (Invitrogen) at 37°C for 1 h, washed, fixed, and permeabilised as described above, and incubated with 500 μl Click-iT reaction cocktail (428 μl Click-iT RNA reaction buffer, 20 μl CuSO$_4$, 1.6 μl Alexa Fluor 488 azide, 50 μl Click-iT reaction buffer additive; Invitrogen) at RT for 30 min in the dark. Cells were washed once in 1 ml Click-iT reaction rinse buffer, sealed, and imaged as described above.

### DNA damage in situ ligation–proximity ligation assay (DI-PLA)

For the DI-PLA, cells were grown, fixed, and permeabilised as above. For DSB blunting, samples were washed twice in 1x CutSmart buffer (2 min, RT; NEB), and incubated (1 h, RT) in 50 μl blunting mix (38.5 μl ddH$_2$O, 5 μl 10x blunting buffer; NEB, 5 μl 1 mM dNTPs, 0.5 μl 20 mg/ml BSA, 1 μl blunting enzyme mix from Quick Blunting Kit; NEB). For ligation, samples were washed twice in 1x CutSmart buffer (2 min, RT; NEB), preincubated (5 min, RT) in 1x T4 ligase buffer (NEB), and incubated (18 h, 16°C with gentle shaking) in 100 μl ligation buffer (83.5 μl ddH$_2$O, 10 μl 10x T4 ligase buffer; NEB, 1 μl 20 mg/ml BSA, 2.5 μl 10 μM biotinylated linker pair [Table S3], 2 μl 2,000 U/$\mu$L T4 ligase; NEB, 1 μl 100 mM ATP solution; NEB). For removal of the excess linker, samples were washed twice in PBS. Proximity ligation assays were performed with a Duolink in situ PLA kit (Sigma-Aldrich) following the manufacturer's protocol and assessed by confocal microscopy as described above. For analysis, data were plotted as % of cells from six individual acquisitions that contain ~50 cells each and were taken as technical replicates.

### Neutral comet assay

Glass slides were covered in ddH$_2$O containing 0.01% poly-L-lysine solution (Sigma-Aldrich) and 1% agarose (Roth) and incubated in a hybridisation oven (UVP) at 70°C overnight. Cells were trypsinised, harvested in 1x PBS, and diluted to 10$^5$ cells/ml. The cell suspension was mixed with an equal volume of 1.5% low melting agarose gel (Biozym) in 1x PBS at 37°C, pipetted on preincubated glass slides, and flattened with a coverslip immediately. The slides were cooled down for 10 min at 4°C. After removal of coverslips, lysis buffer (2.5 M NaCl, 0.1 M EDTA, 0.1 M Tris–HCl, pH 10, 1% Triton x-100) was added directly on the slide, covered with parafilm, and incubated (1 h, 4°C). Slides were washed twice in 1x PBS and analysed by electrophoresis (1 V/cm, 15 min) in neutral comet buffer (100 mM Tris base, pH 8.5, 300 mM sodium acetate) at 4°C. Slides were fixed in 70% EtOH and dried at RT overnight, stained with SYBR Gold

---

**Figure 4. Nono-dependent induction of Gadd45b in KP cells.**
**(A)** Volcano plots displaying RNA-seq data from KP (top) and KPN (bottom) cells. x-axis, relative fold change (FC) in log$_2$ scale; y-axis, *P*-values in −log$_{10}$ scale; threshold, log$_2$ FC > ±1, *P*-value < 0.05. **(B)** RT–qPCR using site-specific primers upon treatment with etoposide. n, number of biological replicates. **(C)** Imaging (top) and quantitation (bottom) of Gadd45b. n, number of cells. **(B, D)** as in (B) upon treatment with bleomycin. */**, *P*-value < 0.05/<0.001; two-tailed *t* test. Error bar, mean ± SD. Source data are available for this figure.

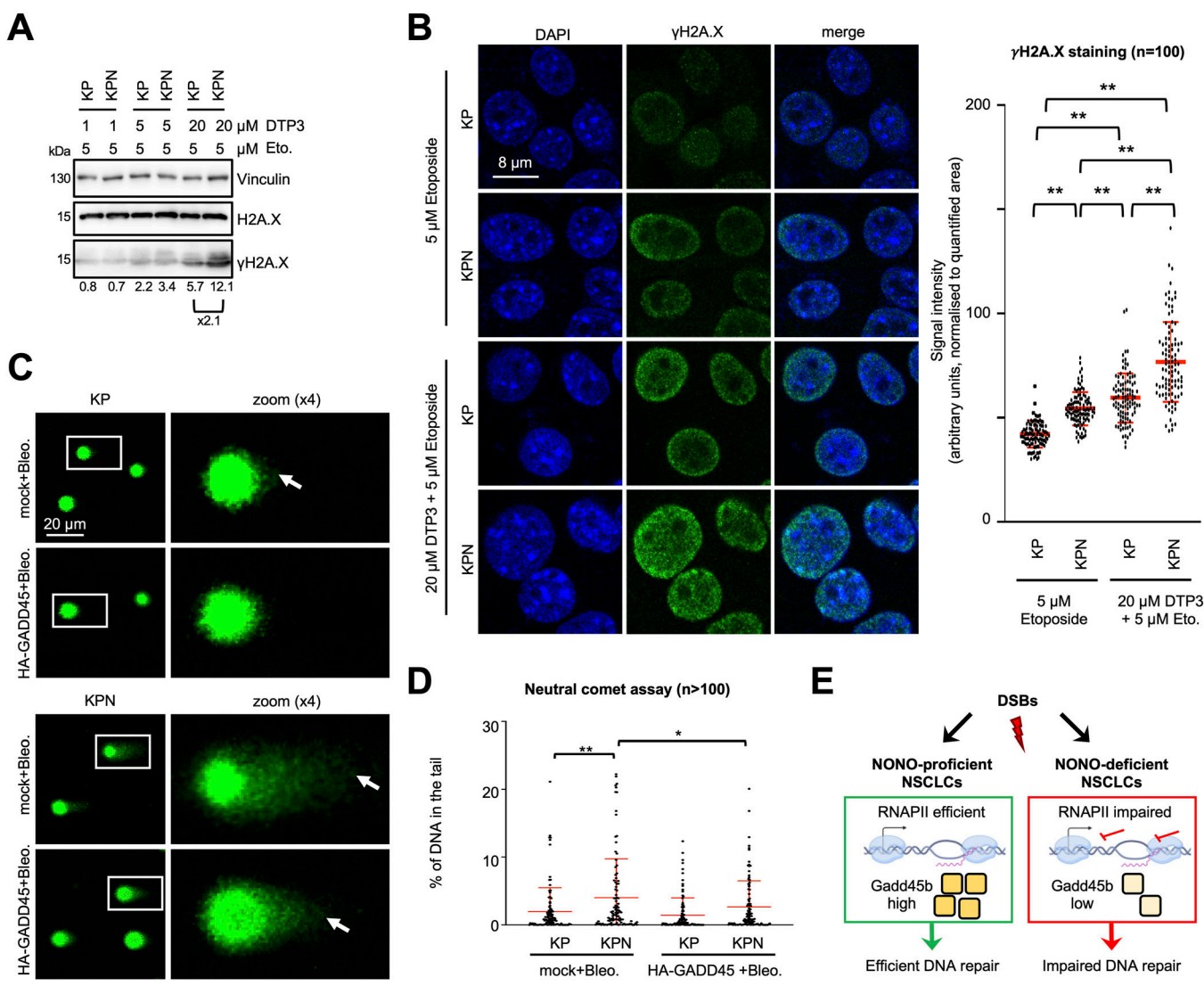

**Figure 5. Gadd45b promotes the DDR in Nono-proficient KP cells.**
**(A, B)** Immunoblots (A) and imaging (B) of γH2A.X. Vinculin and total H2A.X, loading controls; n, number of cells. **(C, D)** Imaging (C) and quantitation (D) of DNA subjected to a neutral comet assay in the absence or presence of ectopically expressed HA-GADD45. White box, zoom; arrowhead, DNA tail; mock; non-transfected control; n, number of cells. **(E)** Model for the Nono-dependent DDR in NSCLC cells. */**, *P*-value < 0.05/<0.001; two-tailed *t* test. Error bar, mean ± SD.
Source data are available for this figure.

(Invitrogen), imaged by confocal microscopy, and quantified using CometScore software.

### ChIP

Cells were fixed with 1% formaldehyde (10 min, 37°C), quenched in 0.125 M glycine (10 min, 37°C), washed in PBS, and centrifuged (320*g*, 5 min). Pellets were resuspended in 500 µl cold cell lysis buffer (5 mM PIPES, pH 8.0, 85 mM KCl, 0.5% NP-40, 1x protease/phosphatase inhibitor) and lysed (10 min on ice). Nuclei were centrifuged (720*g*, 5 min) and resuspended in 300 µl cold nuclear lysis buffer (1% SDS, 10 mM EDTA, 50 mM Tris–HCl, pH 8.0, 1x protease/phosphatase inhibitor) and lysed (10 min on ice). Lysates were sonicated five times (5 min, 30 s on/off) with a Bioruptor

(Diagenode) and pelleted (13,600*g*, 10 min). The supernatant was mixed with 2 ml dilution buffer (0.01% SDS, 1.1% Triton x-100, 1.2 mM EDTA, 16.7 mM Tris–HCl, pH 8.0, 167 mM NaCl, 1x protease/phosphatase inhibitor). Diluted samples were aliquoted, and 5 µg antibodies were added (IP sample) or not (input) and incubated overnight (4°C with rotation). For pull-down, 20 µl of Protein G Dynabeads was added to IP samples, incubated (1.5 h with rotation), immobilised on a magnet, and washed with 800 µl wash buffer A (0.1% SDS, 1% Triton x-100, 2 mM EDTA, 20 mM Tris–HCl, pH 8.0, 150 mM NaCl), buffer B (0.1% SDS, 1% Triton x-100, 2 mM EDTA, 20 mM Tris–HCl, pH 8.0, 500 mM NaCl), buffer C (0.25 M LiCl, 1% NP-40, 1% sodium deoxycholate, 1 mM EDTA, and 10 mM Tris–HCl, pH 8.0), and twice with buffer D (10 mM Tris–HCl, pH 8.0, 1 mM EDTA). For elution, samples were incubated with 500 µl elution buffer (1% SDS, 0.1 M

NaHCO$_3$) for 30 min with rotation. Reversal of crosslinks was performed at 65°C overnight after adding 30 µl 5 M NaCl, 1 µl 10 mg/ml RNase A (Sigma-Aldrich), 10 µl 0.5 M EDTA, 20 µl 1 M Tris–HCl, pH 6.8, 2 µl 10 mg/ml proteinase K (Sigma-Aldrich) to input and IP samples. DNA was purified by phenol/chloroform extraction, recovered in ddH$_2$O, and assessed by quantitative PCR with selective primers (Table S4).

### CUT&RUN sequencing

Cells were harvested with Accutase (Sigma-Aldrich), centrifuged (500$g$, 3 min), and washed three times in 1.5 ml wash buffer (20 mM Hepes, pH 7.5, 150 mM NaCl, 0.5 mM spermidine). Cells were incubated (10 min, RT) with 10 µl concanavalin A–coated magnetic beads (BioMag) resuspended in an equal volume of binding buffer (20 mM Hepes, pH 7.5, 10 mM KCl, 1 mM CaCl$_2$, 1 mM MnCl$_2$), immobilised on a magnet, permeabilised with 150 µl antibody buffer (20 mM Hepes, pH 7.5, 150 mM NaCl, 0.5 mM spermidine, 0.05% digitonin, 2 mM EDTA), and incubated with 1 µg primary antibody (4°C, overnight with rotation). Samples were placed on a magnet, washed two times with 1 ml dig-wash buffer (20 mM Hepes, pH 7.5, 150 mM NaCl, 0.5 mM spermidine, 0.05% digitonin), and incubated (1 h, 4°C with rotation) with 150 µl protein A/G–micrococcal nuclease fusion protein (1 µg/ml; CST). Reactions were placed on a magnet, and washed two times with 1 ml dig-wash buffer and once with 1 ml rinse buffer (20 mM Hepes, pH 7.5, 0.05% digitonin, 0.5 mM spermidine). For chromatin digestion and release, samples were incubated (30 min, on ice) in ice-cold digestion buffer (3.5 mM Hepes, pH 7.5, 10 mM CaCl$_2$, 0.05% digitonin). The reaction was stopped by the addition of 200 µl stop buffer (170 mM NaCl, 20 mM EGTA, 0.05% digitonin, 50 µg/ml RNase A, 25 µg/ml glycogen), and fragments were released by incubation (30 min, 37°C). The supernatant was incubated (1 h, 50°C) with 2 µl 10% SDS and 5 µl proteinase K (10 mg/ml; Sigma-Aldrich). Chromatin was recovered by phenol/chloroform extraction and resuspended in 30 µl TE (1 mM Tris–HCl, pH 8.0, 0.1 mM EDTA). For sequencing, replicates were quantified with a fragment analyser (Advanced Analytical) and subjected to library preparation. Libraries for small DNA fragments (25–75 bp) were prepared based on NEBNext Ultra II DNA Library Prep Kit for Illumina (NEB).

### RNA analytics

Total RNA was isolated using TRIzol (Invitrogen) following the manufacturer's protocol. cDNA was synthesised using SuperScript III reverse transcriptase (Invitrogen) with gene-specific primers (Table S4) and quantified upon reverse transcription–quantitative PCR (RT–qPCR) in a thermocycler (Applied) with PowerUp SYBR Green Master Mix (Applied Biosystems) following the protocols of the manufacturers. For RNA sequencing, cells were directly lysed in 2.1 ml QIAzol (Qiagen). Total RNA was extracted with miRNeasy Kit (Qiagen) using the manufacturer's protocol. 750 ng of total RNA was diluted in 50 µl ddH$_2$O. Individual samples were subjected to a poly(A) mRNA magnetic isolation module (NEB), and libraries were prepared with NEBNext Ultra II Directional RNA Library Prep Kit (NEB) using the manufacturer's protocol. The top 1,000 highly expressed genes in untreated KP cells were used for CUT&RUN-seq

metagene plots. Correlation analysis and visualisation of RNA-seq data were performed using the package "ggplot2" in R software using Spearman's method.

### Generation of FASTQ, BAM, and bedGraph files

For CUT&RUN-seq and RNA-seq, base calling was performed using Illumina's FASTQ generation software v1.0.0 and sequencing quality was tested using FastQC. Reads were mapped with STAR v2.7.10a (RNA-seq) or Bowtie2 v2.3.5.1 (CUT&RUN-seq) to the murine genome mm10, and normalised to the number of mapped reads. BAM files obtained after normalisation were sorted and indexed using SAMtools v1.9. bedGraph files were generated using the genomecov function from BEDTools v2.26.0 (Quinlan & Hall, 2010). CUT&RUN-seq metagene plots of the top 1,000 highly expressed genes were generated using the R package "metagene" with the assay parameters "ChIPseq" and 100 bins. The Integrated Genome Browser was used to visualise these density files. The CUT&RUN-seq signal sum count was performed using BEDTools intersect (Quinlan & Hall, 2010) and visualised with RStudio.

### Generation of density and volcano plots

For RNA-seq, gene expression was assessed with featureCounts v2.0.3 using bam files and similarity between replicates of each condition was verified by PCA plot v2.2.1 (Galaxy). Differential gene expression was assessed with edgeR v3.24.1 (Galaxy) using the Benjamini–Hochberg adjusted $P$-value < 0.05, and an expression filter excluding non- and weakly expressed genes. Genes were plotted with Volcano plot tool v0.05 (Galaxy). Genes with an FDR $P$-value < 0.05 and log$_2$FC > 1 or less than –1 were considered significantly up-regulated or down-regulated, respectively. GSEA between conditions was performed with GSEA-Broad tool v4.1.0 (Galaxy), and only gene sets with an FDR < 25% were considered.

## Data Availability

Sequencing data are available at the Gene Expression Omnibus under the accession number GEO: GSE250302. Further information and requests for resources and reagents should be directed to and will be fulfilled by the corresponding author.

## Supplementary Information

## Acknowledgements

We acknowledge Martin Eilers, Markus Diefenbacher, and Elmar Wolf for feedback and discussions. We thank Peter Gallant, Nikolett Pahor and Oliver Hartmann for technical support and the department for sharing reagents and collegial atmosphere. This work was supported by the German Cancer

Aid (the Dr. Mildred Scheel Stiftung für Krebsforschung, Mildred-Scheel-Nachwuchszentrum, MSNZ, grant number 8606100-NG1) awarded to K Burger and the Open Access Publication Fund of the University of Würzburg.

## Author Contributions

V Mamontova: formal analysis, investigation, methodology, and writing—original draft.
B Trifault: formal analysis, investigation, methodology, and writing—original draft.
K Burger: conceptualisation, formal analysis, supervision, funding acquisition, investigation, methodology, and writing—original draft, review, and editing.

## Conflict of Interest Statement

The authors declare that they have no conflict of interest.

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
