## [Reviewer comments · Life Science Alliance]

Nono induces Gadd45b to mediate DNA repair

Victoria Mamontova, Barbara Trifault, and Kaspar Burger

DOI: <https://doi.org/10.26508/lsa.202302555>

Corresponding author(s): *Kaspar Burger, University Hospital Wuerzburg*

Review Timeline:

Submission Date:	2023-12-23
Editorial Decision:	2024-03-11
Revision Received:	2024-05-16
Editorial Decision:	2024-05-22
Revision Received:	2024-05-23
Accepted:	2024-05-23

Scientific Editor: *Eric Sawey, PhD*

Transaction Report:

March 11, 2024

Re: Life Science Alliance manuscript #LSA-2023-02555-T

Dr. Kaspar Burger
University Hospital Würzburg
Josef-Schneider-Str. 2
Würzburg 97080
Germany

Dear Dr. Burger,

Thank you for submitting your manuscript entitled "Nono stimulates the DNA damage response by mediating the induction of Gadd45b" to Life Science Alliance. The manuscript was assessed by expert reviewers, whose comments are appended to this letter. We invite you to submit a revised manuscript addressing the Reviewer comments.

Thank you for this interesting contribution to Life Science Alliance. We are looking forward to receiving your revised manuscript.

Sincerely,

B. MANUSCRIPT ORGANIZATION AND FORMATTING:

Reviewer #1 (Comments to the Authors (Required)):

Mamontova et al manuscript, provides additional evidences for a role of the RNA-binding protein NONO, in DNA repair end characterize a novel link between NONO function in maintaining genome stability and Gadd45b expression. The author observe that upon etoposide NONO controls the occupancy of RNA Polymerase II with phosphorylated CTD on S5, at promoter of DNA damage responsive genes and link the lack of NONO with accumulation of DNA damage and activation of DDR in NSCLC cells KO for NONO. Also, they propose an inverse correlation between R-loops accumulation upon Etoposide treatment and NONO expression. In the final part of the study, they propose that R-loops are in fact resolved by Gadd45b, e DNA damage responsive gene, who's expression in induces upon etoposide only in NONO expressing cells. The work is well written and interesting despite its novelty is partial since NONO has been shown to play a role in DNA repair, by controlling LLPS. Overall I support the publication of this study in LSA after a minor revision.

Specific points:

- 1) All the study is made in cells treated with Etoposide. This might be to characterize the impact of a DNA damaging agent used in chemotherapy. But to understand the mechanism of action of NONO in DNA damage response and DNA repair it would be useful to test if crucial observations are also reproduced in cells damaged by other DNA damaging agents such as cisplatin or neocarzinostatin, a radiomimetic drug, thus easy to use in cells in culture. This would address if NONO affects expression of DNA damage responsive genes upon different type of DNA damage.
- 2) When discussing figure 3 the authors claim that the lack of NONO impacts on DNA damage signaling. In my opinion this is not correct. What they observe with this experiment is a good signaling since gH2AX accumulates and 53BP1 foci too. Nevertheless, in the absence of NONO they clearly detect a defect in DNA repair as demonstrated by the persistence of DNA damage (gH2AX signal) for longer than control cells and for the previous data obtained by comet.
- 3) To prove the role of Gadd45b in counteracting R-loops would be important to perform the same experiment of panel 1 E and in figure 2 in cells defective for NONO and overexpressing Gadd45b. Possibly, in parallel would be interesting to show an increase of S9.6 signal in control cell treated with Gadd45b inhibitor. This would support the model they propose in the graphical abstract in panel Figure 4G in which DNA damage accumulation in the absence of NONO or Gadd45b is due to R-loops accumulation. Impaired DDR again is not correct. I would ask to put Impaired DNA repair.

For the same reason the running title better fits the data of the study.

Minor point:

Graph of Figure 2B is presented in a counterintuitive fashion. Why start showing less cells with 0 DI-PLA signal upon ETO e ETO+NONO inactivation. Since at the end they want to stress more DNA damage in cells without NONO, would be more clear firstly to show an increase in the % of cells with 1, 2 o 3 spots (possibly also all together). It takes a wile to understand what they want to say with that graph. One could also use bar plots with two colors indicating positive and negative cells on the same bar.

In addition Y axis says: each dot integrate data from 1 acquisition. What is the meaning f 1 acquisition? An independent experiment? A technical replicate, multiple acquisition of the same experiment? This should be better indicated.

Reviewer #2 (Comments to the Authors (Required)):

This study analyzes the effect of NONO, an RNA binding protein, on the DNA damage response using a murine cell-based lung cancer model (KP (KRasG12D, Trp53-/-)). Loss of NONO is shown to induce DNA damage, potentially through R-loops. RNA-Seq data identifies several genes that are regulated by NONO, including after etoposide. One of these genes is Gadd45b, which the author provide supporting data using an inhibitor which displays similar DNA damage phenotypes compared to NONO-deficient cells. Some cut and run data is provided that is consistent with NONO regulating this gene. Overall, the scope of this work is limited and the data is not of sufficient quality or depth to make strong conclusions. The significance of these findings is

limited with the presented data and it is unclear how robust and universal these findings are. Given these issues (and those provided in more detail below), this work is too preliminary for publication in its current form.

Main issues

1. The data in Figure 1E is of poor quality. The dot blots do not seem to be performed at a level where clear conclusions can be made. The loading shows rings and is not equal for the SYBR detected samples. The S9.6 signal are low and don't seem to represent the dilutions being used. For example, the difference between 500 and 250 ng does not seem to be half. Given these data, strong conclusions can't be made. To support the idea that global R-loops increase under these conditions, additional supportive and more convincing data needs to be obtained. In addition, it would be informative to show samples without Eto for R-loops to see if there are any changes that occur between KP and KPN for R-loops. These additional data are important to allow for damage-dependent changes to be interpreted.
2. The DNA damage effects observed in Figure 2 are very minor. This may be due to the level of damage induced by Etoposide with the concentration and time frame that is used. Can the authors perform these experiments where more breaks are observed. This may show an even more robust different in NONO-deficient cells.
3. The western blot in Figure 4C is of poor quality, making any conclusions difficult to make. Quantifying western blots is notoriously unreliable. This data needs to be reproduced with better loading and gel quality to show this effect more convincingly.
4. Figure 4E requires a H2AX unmodified total blot for a loading control.
5. The authors observe cut and run data that is consistent with Gadd45b being regulated by NONO. This analysis should be done with the entire set of upregulated genes observed in the RNA-Seq data to see if this is observed with all NONO-regulated genes.
6. How is the specificity of the Gadd45b inhibitor assessed? It would be informative to perform a siRNA knockdown of Gadd45b to support the data shown with the inhibitor (i.e. Figure 4E).
7. Are these results specific to lung cancer? Testing in multiple cell lines, including different cancers could address this important question.
8. Does overexpression of RNaseH suppress the DNA damage that is observed in NONO-deficient cells?

We thank the editor and reviewers for the opportunity to submit a revised version of our manuscript. We also thank them for their valuable suggestions, which have substantially increased the quality of the manuscript. We here provide point-by-point answers to all concerns raised by reviewers. We are confident that our revised manuscript has improved significantly and we hope that you will appreciate the improvements.

RESPONSE TO REVIEWER 1

Reviewer #1 (Comments to the Authors (Required)):

Mamontova et al manuscript, provides additional evidences for a role of the RNA-binding protein NONO, in DNA repair end characterize a novel link between NONO function in maintaining genome stability and Gadd45b expression. The author observe that upon etoposide NONO controls the occupancy of RNA Polymerase II with phosphorylated CTD on S5, at promoter of DNA damage responsive genes and link the lack of NONO with accumulation of DNA damage and activation of DDR in NSCLC cells KO for NONO. Also, they propose an inverse correlation between R-loops accumulation upon Etoposide treatment and NONO expression. In the final part of the study, they propose that R-loops are in fact resolved by Gadd45b, e DNA damage responsive gene, who's expression in induces upon etoposide only in NONO expressing cells. The work is well written and interesting despite its novelty is partial since NONO has been shown to play a role in DNA repair, by controlling LLPS.

Overall I support the publication of this study in LSA after a minor revision.

Specific points:

1) All the study is made in cells treated with Etoposide. This might be to characterize the impact of a DNA damaging agent used in chemotherapy. But to understand the mechanism of action of NONO in DNA damage response and DNA repair it would be useful to test if crucial observations are also reproduced in cells damaged by other DNA damaging agents such as cisplatin or neocarzinostatin, a radiomimetic drug, thus easy to use in cells in culture. This would address if NONO affects expression of DNA damage responsive genes upon different type of DNA damage.

We agree that it is useful to reproduce crucial observations by using additional DNA-damaging agents. Following the suggestion of reviewer 1, we chose the radiomimetic drug Bleomycin and the DNA interstrand-crosslinker Cisplatin. Supporting a role for Nono in the recognition and repair of DSBs, Nono-deficient cells were hypersensitive to treatment with DSB-inducing Bleomycin (as measured by the neutral comet assay) and displayed elevated levels of gH2AX upon pulse-chase treatment with bleomycin (as measured by immunoblotting). We could also confirm the attenuated induction of Gadd45b expression in Nono-deficient cells both upon etoposide and bleomycin treatment by RT-qPCR and confocal imaging.

Of note, we initially used etoposide in our assays for two main reasons: (i) etoposide is widely used for the treatment of various malignancies including lung cancer (Souhami et al., JNCI 1997) and (ii) our recently published data suggest that human NONO mediates the repair of non-persistent, nucleoplasmic DSBs, which seem to be the predominant type of lesions upon moderate treatment with etoposide (Trifault et al., 2024 NAR; Hornofova et al., 2023 eLife). Etoposide acts as a pure TOP2 poison that generates relatively 'clean and easy-to-repair' DSBs whilst other topoisomerase inhibitors are more complex in their modes of action (Cortes Ledesma et al. 2009 Nature).

2) When discussing figure 3 the authors claim that the lack of NONO impacts on DNA damage signaling. In my opinion this is not correct. What they observe with this experiment is a good signaling since gH2AX accumulates and 53BP1 foci too. Nevertheless, in the absence of NONO they clearly detect a defect in DNA repair as demonstrated by the persistence of DNA damage (gH2AX signal) for longer than control cells and for the previous data obtained by comet.

We agree and have changed the wording accordingly.

3) To prove the role of Gadd45b in counteracting R-loops would be important to perform the same experiment of panel 1 E and in figure 2 in cells defective for NONO and overexpressing Gadd45b. Possibly, in parallel would be interesting to show an increase of S9.6 signal in control cell treated with Gadd45b inhibitor. This would support the model they propose in the graphical abstract in panel Figure 4G in which DNA damage accumulation in the absence of NONO or Gadd45b is due to R-loops accumulation.

We agree that additional data are important to convincingly demonstrate the suggested role of Gadd45b in counteracting R-loops. In view of the comments of reviewer 2, who raised concerns on the quality and depth of our R-loop data (in particular Figure 1E), we no longer propose an inverse correlation between R-loops accumulation upon Etoposide treatment and Nono/Gadd45b expression. We adjusted our model accordingly. We now solely claim that Nono promotes genome stability by promoting the efficient induction of the bona fide DDR factor Gadd45b. We have added new data that support the link between Nono-mediated DDR and Gadd45b. We discuss that additional functions for Nono in the DDR exist, which likely includes the suppression R-loops (Trifault et al., 2024 NAR; Petti et al., 2019), which may further contribute to Nono's genome-protective role in KP cells.

Impaired DDR again is not correct. I would ask to put Impaired DNA repair. For the same reason the running title better fits the data of the study.

We agree and have adjusted title and running title accordingly.

Minor point:

Graph of Figure 2B is presented in a counterintuitive fashion. Why start showing less cells with 0 DI-PLA signal upon ETO e ETO+NONO inactivation. Since at the end they want to stress more DNA damage in cells without NONO, would be more clear firstly to show an increase in the % of cells with 1, 2 o 3 spots (possibly also all together). It takes a wile to understand what they want to say with that graph. One could also use bar plots with two colors indicating positive and negative cells on the same bar.

We apologise for the confusion. We have changed the order of columns, but would request to keep the rest of the presentation of panel 2B as suggested in the initial data set.

In addition Y axis says: each dot integrate data from 1 acquisition. What is the meaning f 1 acquisition? An independent experiment? A technical replicate, multiple acquisition of the same experiment? This should be better indicated.

We again apologise for the confusion. One acquisition means one image taken on the microscope. Each image contains 50 or so cells, 6 images were taken as technical replicates and the total number of cells analysed (n) equals 300. We have adjusted the methods section accordingly.

Overall, we hope that our adjustments sufficiently improve the quality or depth of the data set.

We thank the editor and reviewers for the opportunity to submit a revised version of our manuscript. We also thank them for their valuable suggestions, which have substantially increased the quality of the manuscript. We here provide point-by-point answers to all concerns raised by reviewers. We are confident that our revised manuscript has improved significantly and we hope that you will appreciate the improvements.

RESPONSE TO REVIEWER 2

Reviewer #2 (Comments to the Authors (Required)):

This study analyzes the effect of NONO, an RNA binding protein, on the DNA damage response using a murine cell-based lung cancer model (KP (KRasG12D, Trp53^{-/-}). Loss of NONO is shown to induce DNA damage, potentially through R-loops. RNA-Seq data identifies several genes that are regulated by NONO, including after etoposide. One of these genes is Gadd45b, which the author provide supporting data using an inhibitor which displays similar DNA damage phenotypes compared to NONO-deficient cells. Some cut and run data is provided that is consistent with NONO regulating this gene. Overall, the scope of this work is limited and the data is not of sufficient quality or depth to make strong conclusions. The significance of these findings is limited with the presented data and it is unclear how robust and universal these findings are. Given these issues (and those provided in more detail below), this work is too preliminary for publication in its current form.

Main issues

1. The data in Figure 1E is of poor quality. The dot blots do not seem to be performed at a level where clear conclusions can be made. The loading shows rings and is not equal for the SYBR detected samples. The S9.6 signal are low and don't seem to represent the dilutions being used. For example, the difference between 500 and 250 ng does not seem to be half. Given these data, strong conclusions can't be made. To support the idea that global R-loops increase under these conditions, additional supportive and more convincing data needs to be obtained. In addition, it would be informative to show samples without Eto for R-loops to see if there are any changes that occur between KP and KPN for R-loops. These additional data are important to allow for damage-dependent changes to be interpreted.

We agree that additional data are important to convincingly demonstrate a link to R-loops. In view of the comments of reviewer 1, who also raised some concerns on our R-loop data, we no longer propose an inverse correlation between R-loops accumulation upon Etoposide treatment and Nono/Gadd45b expression. We now solely claim that Nono promotes genome stability by promoting the efficient induction of the bona fide DDR factor Gadd45b. We have added new data that support the link between Nono-mediated DDR and Gadd45b. We discuss that additional functions for Nono in the DDR exist, which likely includes the suppression R-loops (Trifault et al.,

2024 NAR; Petti et al., 2019), which may further contribute to Nono's genome-protective role in KP cells.

2. The DNA damage effects observed in Figure 2 are very minor. This may be due to the level of damage induced by Etoposide with the concentration and time frame that is used. Can the authors perform these experiments where more breaks are observed. This may show an even more robust difference in NONO-deficient cells.

We agree that a more robust phenotype in figure 2 is desirable. Therefore, we increased both the concentration and the incubation time of etoposide (50 μ M, 4h), and included two additional DNA-damaging agents (the radiomimetic, DSB-inducing drug Bleomycin and the DNA interstrand-crosslinker Cisplatin) in the analysis (which has also been suggested by reviewer 1). Supporting a role for Nono in the recognition and repair of DSBs, Nono-deficient cells were hypersensitive to treatment with Bleomycin as measured by the neutral comet assay. Surprisingly, treatment with 50 μ M of etoposide for 4h did not significantly improve the initial comet phenotype. However, the excessive DNA damage phenotype upon combining bleomycin treatment with Nono deletion could be partially rescued by the overexpression of HA-GADD45 in KPN cells.

In our opinion, the observed DI-PLA phenotype is in line with published observations, which vary in terms of DI-PLA foci number depending on the cell types/organisms assessed (compare for example different DI-PLA foci number between mouse cell types in Fig 2a and Fig 2c in Galbiati et al., 2014 Aging Cell). The rather minor number of DI-PLA foci in our study may indeed be due to the relatively modest induction of DSBs by etoposide. Etoposide acts as a pure TOP2 poison that generates relatively 'clean and easy-to-repair' DSBs whilst other topoisomerase inhibitors are more complex in their modes of action (Cortes Ledesma et al. 2009 Nature). Nevertheless, the concentration and time frame of etoposide treatment used in this study (20 μ M, 2h +chase) clearly induces markers of DNA damage (see γ H2AX and 53BP1 data Fig 3 and the RNA-seq data from Figure 4). The latter further suggests an induction of several bona fide DNA damage-responsive transcripts (CDKN1A, GADD45B, PLK2, FOSB) at the concentration and time frame of etoposide treatment used in this study. Moreover, using 50 μ M etoposide for 4h only modestly improved our comet assay phenotype, which indicates that KP/KPM cells may intrinsically tolerate moderate etoposide treatment. Moreover, DI-PLA data were not obtained from an epifluorescence microscope. They are displayed as snap shots of a single confocal section (and not multiple aggregated Z-stacks). Our DI-PLA phenotype may appear weak for any of those reasons, nevertheless, it is significant. Of note, we have changed the order of columns, as suggested by reviewer 1

3. The western blot in Figure 4C is of poor quality, making any conclusions difficult to make. Quantifying western blots is notoriously unreliable. This data needs to be reproduced with better loading and gel quality to show this effect more convincingly.

We used a highly sensitive, state of the art device for imaging of Gadd45b ECL chemiluminescence (Vilber Fusion FX, which can detect signals in the linear range from very low ECL signal intensities). Nevertheless, we agree that the initial immunoblots are of poor quality. We have repeated the experiment, which improved the quality of the Vinculin blot, and confirmed the Gadd45b phenotype (albeit with again rather weak reactivity of Gadd45b antibody). We therefore moved the immunoblot data to Supplementary Information.

To demonstrate the phenotype more convincingly, we used the Gadd45b antibody in confocal imaging and could indeed confirm attenuated induction of Gadd45b in KPN cells both upon etoposide and bleomycin treatment.

4. Figure 4E requires a H2AX unmodified total blot for a loading control.

We agree and have added the H2A.X unmodified total blot as additional loading control to confirm the finding.

5. The authors observe cut and run data that is consistent with Gadd45b being regulated by NONO. This analysis should be done with the entire set of upregulated genes observed in the RNA-Seq data to see if this is observed with all NONO-regulated genes.

We agree and have done the analysis for a total number of 10 genes that are all upregulated by etoposide treatment in KP cells. The induction of 5 of them (group A, all part of TNFalpha signalling gene set, including Gadd45b) was sensitive to Nono knockout, the other 5 (group B) were still induced in KPN cells. Strikingly, RNAPII CTD S5P occupancy in KPN cells was sensitive to etoposide treatment at the promoters of 4 of the 5 members of group A, but not any member of group B, further suggesting that Nono selectively stimulates the induction of group A genes upon etoposide treatment.

Of note, we did not analyse the entire set of all 162 genes that were upregulated by etoposide in KP cells, as we wished to maintain a balanced and manageable amount of browser tracks within the data set. Furthermore, we do not claim that Nono regulates those genes solely at the transcriptional level, but also by other mechanisms as discussed in the manuscript.

6. How is the specificity of the Gadd45b inhibitor assessed? It would be informative to perform a siRNA knockdown of Gadd45b to support the data shown with the inhibitor (i.e. Figure 4E).

We thank the reviewer for this important comment. Gadd45b binds the stress-responsive MAP kinase kinase MKK7 to impair its activity. The interaction of DTP3 with MKK7 prevents the binding of Gadd45b to MKK7, which activates MKK7 and triggers phosphorylation of JNK kinase and efficient MAP kinase signalling (Tornatore et al., 2014 Cancer Cell; Sandomenico et al., 2021 Biomedicines). Thus, we assessed the impact of DTP3 treatment on DNA-damage induced JNK activation by immunoblotting. We found that treatment with a high concentration of DTP3 (20 μ M)

correlates with high levels of phospho-JNK in both KP and KPN cells, in the presence of etoposide. This suggests that DTP3 is active in both KP and KPN cells. A more detailed analysis of DTP3 specificity requires more sophisticated biochemical assays (as performed in Tornatore et al., 2014 Cancer Cell) and may be beyond the scope of this manuscript.

Of note, we also tried to deplete Gadd45b by RNAi using a commercially available mouse Gadd45b shRNA plasmid kit (abx971640, abbexa). The kit contains three plasmids that encode three different shRNAs and a GFP reporter to estimate transfection efficacy. Unfortunately, this approach suffers from low transfection efficacies (less than 10%) (see the rebuttal figure). We assume that this may perhaps be due to the tendency of KP/KPN cells to form cellular clumps upon incubation with OptiMEM and transfection stress. As low transfection efficacy and variation in shRNA expression levels likely translate into a heterogeneous population of KP/KPN cells with poor Gadd45b knockdown efficacy, we argue that the knockdown approach suggested by reviewer 2 is not feasible in the KP/KPN system. We ask to omit this experiment and instead acknowledge the HA-GADD45 overexpression experiment, which support the data shown with the inhibitor.

Rebuttal figure

KP cells transfected with pGPU6/GFP/Neo plasmid (abx971640, abbexa) that co-expresses Gadd45b-targeting shRNA and GFP

7. Are these results specific to lung cancer? Testing in multiple cell lines, including different cancers could address this important question.

We are aware that the novelty of the data may appear limited and that the conclusions drawn from it may be somewhat limited to the NSCLC system used in this study, as indicated in the model and throughout the manuscript. We are confident that the

revised data set has improved in robustness and is now of sufficient depth and significance for the scope of Life Science Alliance. In our humble opinion, testing our model in different cancers and multiple cell lines is beyond the scope of this study and may be subject to future studies in vivo.

8. Does overexpression of RNaseH suppress the DNA damage that is observed in NONO-deficient cells?

We agree that this experiment would strengthen our initial conclusions. As we no longer propose an inverse correlation between R-loops accumulation upon Etoposide treatment and Nono/Gadd45b expression, we would kindly ask to omit this experiment.

Overall, we hope that our adjustments sufficiently improve the quality or depth of the data set.

May 22, 2024

RE: Life Science Alliance Manuscript #LSA-2023-02555-TR

Dr. Kaspar Burger
University Hospital Würzburg
Josef-Schneider-Str. 2
Würzburg 97080
Germany

Dear Dr. Burger,

Thank you for submitting your revised manuscript entitled "Nono induces Gadd45b to mediate DNA repair". We would be happy to publish your paper in Life Science Alliance pending final revisions necessary to meet our formatting guidelines.

- please be sure that the authorship listing and order is correct
- please upload your Tables in editable .doc or excel format
- please add the Twitter handle of your host institute/organization as well as your own or/and one of the authors in our system
- please use the [10 author names et al.] format in your references (i.e., limit the author names to the first 10)
- please add your main, supplementary figure, and table legends to the main manuscript text after the references section
- please remove legends from the supplementary figures. Their captions should appear only in the manuscript file.

A. FINAL FILES:

B. MANUSCRIPT ORGANIZATION AND FORMATTING:

**Submission of a paper that does not conform to Life Science Alliance guidelines will delay the acceptance of your

manuscript.**

The license to publish form must be signed before your manuscript can be sent to production. A link to the electronic license to publish form will be available to the corresponding author only. Please take a moment to check your funder requirements.

Sincerely,

May 23, 2024

RE: Life Science Alliance Manuscript #LSA-2023-02555-TRR

Dr. Kaspar Burger
University Hospital Würzburg
University of Würzburg, LS Biochem u. Mol.biol
Am Hubland
MSNZ, AG Burger, Biozentrum
Würzburg 97080
Germany

Dear Dr. Burger,

Thank you for submitting your Research Article entitled "Nono induces Gadd45b to mediate DNA repair". It is a pleasure to let you know that your manuscript is now accepted for publication in Life Science Alliance. Congratulations on this interesting work.

DISTRIBUTION OF MATERIALS:

Again, congratulations on a very nice paper. I hope you found the review process to be constructive and are pleased with how the manuscript was handled editorially. We look forward to future exciting submissions from your lab.

Sincerely,
